# Determinants of stunting in children under five years old in South Sulawesi and West Sulawesi Province: 2013 and 2018 Indonesian Basic Health Survey

**Hayani Anastasia**[1]*, **Veni Hadju**[2], **Rudy Hartono**[3], **Samarang**[1], **Manjilala**[3], **Sirajuddin**[3], **Abdul Salam**[2], **Atmarita**[4]

1 Research Center for Public Health and Nutrition, Research Organization for Health, National Research and Innovation Agency, Republic of Indonesia, Jakarta, Indonesia, 2 Department of Nutrition, School of Public Health, Hasanuddin University, Makassar, Indonesia, 3 Department of Nutrition, Makassar Health Polytechnic, Makassar, Indonesia, 4 Indonesia Health Researcher Association, Jakarta, Indonesia

* anayani7337@gmail.com

**Data Availability Statement:** The 2013 and 2018 Indonesian Basic Health Survey data used to support these study findings were supplied by the

## Abstract

### Background

The prevalence of stunting in South Sulawesi and West Sulawesi Province is relatively high. Studies reveal various household, parental, and child factors are associated with stunting. This paper aimed to determine the determinants of stunting among children under five in South and West Sulawesi Province using the 2013 and 2018 Indonesia Basic Health Survey.

### Methods

This study was a secondary data analysis using the 2013 and 2018 Indonesian Basic Health Surveys. The unit of analysis was children under five years, and the study obtained 3641 and 4423 children in South Sulawesi Province from the 2013 and 2018 Indonesia Basic Health Survey, respectively, and 804 and 1059 children from the 2013 and 2018 Indonesia Basic Health Survey in West Sulawesi Province, respectively. The multivariable poisson regression model was conducted to determine the determinants of stunting.

### Results

The results showed that the mean age of children in South Sulawesi and West Sulawesi Province was 31.1 months and 30.8 months, respectively, on 2013 survey compare to 29.6 months and 29.1 months on the 2018 survey. The determinants of stunting in children under five in South Sulawesi Province in 2013 survey were household with two children under five (APR:1.24; 95% CI: 1.1–1.3; p-value = 0.006), maternal BMI (APR: 1.1; 95% CI: 1.03–1.1; p-value = <0.001), maternal weight (APR: 0.9; 95% CI: 0.94–0.97; p-value = <0.001), children aged 12–23 months (APR: 2.2; 95% CI: 1.7–2.7; p-value = <0.001), children aged 24–59 months (APR: 2.2; 95% CI: 1.8–2.7; p-value = <0.001), birthweight (APR: 1.9; 95% CI:

Health Development Policy Agency of the Indonesian Ministry of Health under license. Requests for access to these data should be made to the Health Development Policy Agency of the Indonesian Ministry of Health (http://www.badankebijakan.kemkes.go.id/layanan-permintaan-data/; Email: datin.bkpk@kemkes.go.id).

**Funding:** This study was supported by grants from Asian Development Bank (ADB) and Ministry of Health for data management, data analysis, manuscript writing, and publication (Number: 046/MADEP-ADB/KONTRAK/III/2022). Authors received the Award: HA, VH, RH, Sa, M, Si, and AS. The funders had no role in study design, data collection and analysis, decision to publish, or preparation of the manuscript.

**Competing interests:** The authors have declared that no competing interests exist.

1.5–2.5; p-value = <0.001). In the 2018 survey, the determinants were maternal weight (APR: 0.9; 95% CI: 0.98–0.99; p-value = 0.005), mothers with no education or with education in primary school (APR: 1.5; 95% CI: 1.3–1.9; p-value = <0.001), mothers with education in middle school (APR: 1.3; 95% CI: 1.1–1.6; p-value = 0.014), mothers with a height less than 151 cm (APR: 1.3; 95% CI: 1.5–3.1; p-value = <0.001), mothers with a height of 151–160 cm (APR: 1.3; 95% CI: 1.1–1.6; p-value = 0.014), children aged 12–23 months (APR: 2.2; 95% CI: 1.7–2.8; p-value = <0.001), children aged 24–59 months (APR: 2.5; 95% CI: 2.0–3.2; p-value = <0.001)., birthweight (APR: 1.5; 95% CI: 1.2–1.9; p-value = <0.001). The determinants of stunting in children under five in West Sulawesi Province in 2013 survey were children under five years living in poor households (APR: 1.9; 95% CI: 1.1–3.3; p-value = 0.021), children under five who lived in a household with three or more children under five (APR:1.8; 95% CI: 1.2–2.7; p-value = 0.002), children aged 12–23 months (APR: 1.8; 95% CI: 1.2–2.6; p-value = 0.006), children aged 24–59 months (APR: 1.9; 95% CI: 1.3–2.7; p-value = 0.001). On the 2018 survey, the determinants were maternal BMI (APR: 1.02; 95% CI: 1.01–1.03; p-value = 0.004), maternal weight (APR: 0.9; 95% CI: 0.95–0.98; p-value = <0.001), mothers with no education or with education in primary school (APR: 1.9; 95% CI: 1.3–2.7; p-value = 0.001), mothers with education in middle school (APR: 1.9; 95% CI: 1.3–2.7; p-value = 0.001), mothers with education in high school (APR: 1.8; 95% CI: 1.2–2.6; p-value = 0.004), children aged 12–23 months (APR: 2.1; 95% CI: 1.4–3.0; p-value = <0.001), children aged 24–59 months (APR: 2.6; 95% CI: 1.9–3.6; p-value = <0.001), male (APR: 1.2; 95% CI: 1.01–1.4; p-value = 0.035), and acute respiratory infection (ARI) (APR: 1.6; 95% CI: 1.04–2.5; p-value = 0.030).

## Conclusions

The determinants of stunting in children under five in South Sulawesi Province are number of children under five in the household, maternal education, maternal weight, maternal height, maternal BMI, child's age, and birthweight. Meanwhile, the determinants of stunting in children under five in West Sulawesi were wealth, maternal education, maternal weight, maternal BMI,, child's age, child's sex, and history of acute respiratory infection. Hence, intervention on household and child levels as well as maternal sociodemographic factors need to be addressed.

## Introduction

Stunting remains a global public health issue, particularly in low and middle-income countries. UNICEF/WHO and the World Bank estimate that approximately 151 million children are stunted, or about 22.2% of children worldwide, where more than half are from Asia [1]. Stunting has long-lasting consequences on cognitive ability, school performance, and socioeconomic status [2]. A stunted child can never reach his/her optimal height and have poor cognitive development [1, 3–5].

Stunting in Indonesia remains high, with a 30–39% prevalence according to WHO's cut-off values for public health significance for stunting [5]. The World Bank (2020) reported that Indonesia has underperformed in reducing the level of stunting compared to other upper-middle-income countries and other countries in the region [6]. The reduction of stunting in

children under five years old was only 10% (a decrease from 46.8% to 36.7%) based on the Indonesia Family Survey from 1997–2007 [7]. This finding is in line with the 2013 Basic Health Survey that reported that36.8% children in Indonesia were stunted [8].

Studies reveal various household, parental, and child factors are associated with stunting. Study in Nepal, Ethiopia, Bangladesh reported that household factors like socioeconomic status, household size, and the number of children in the family are related to stunting. Studies have reported that children from higher socioeconomic strata have a lower risk of stunting than those from lower socioeconomic strata [9–11]. Parental factors, such as education and the mother's height, are correlated with stunting as reported in Ethiopia, Bangladesh, and Pakistan [10–12]. Biological factors include infections (like diarrhea), low birth weight, breastfed, child's sex, and age [13, 14]. The prevalence of stunting varies in Indonesia. In 2018, Jakarta had the lowest prevalence with 17.7%, and East Nusa Tenggara had the highest prevalence with 42.6% [15]. The eastern part of Indonesia has the highest prevalence of stunting because many development indicators are behind other regions [6]. The disparity in stunting between districts in South Sulawesi and West Sulawesi (Sulselbar) is relatively high. Based on the 2021 survey, some communities that are still high in South Sulawesi are Jeneponto (37.9%), Maros (37.5%), and Takalar (34.7%). In Enrekang District, the prevalence of stunting decreased from 42.3% in 2018 to 31.7% in 2021. However, the decline at the sub-district level varied where prevalence was still high in IDD (Iodine Deficiency Disorders) endemic sub-districts. As for West Sulawesi Province, several districts are still high, including Polman Regency (36.0%), Majene (35.7%), and Mamasa (33.7%). These three areas are known as IDD endemic areas [16].

This paper examines the household, maternal, and individual determinants of stunting in South and West Sulawesi using data from the 2013 and 2018 Indonesian Health Basic Survey. This paper aimed to determine the determinants of stunting among children under five in South and West Sulawesi. This remaining article proceeds as follows: section 2 discuss materials and methods, section 3 and 4 discuss result and discussion, and the last section describes this paper's conclusion.

## Materials and methods

### Data source and survey design

The data used in this analysis is secondary data from 2013 and 2018 Indonesian Basic Health Surveys conducted by the National Institute of Health Research and Development, Ministry of Health, Republic of Indonesia. It is a cross-sectional survey conducted every five years to portray the health situation of the community at the city/district, province, and national levels.

The 2013 Indonesia Basic Health Survey survey included households from all 33 provinces and 497 districts/cities of Indonesia. The survey used a multi-stage sampling with 11,986 census blocks visited of 12,000 census blocks targeted (99.9%); 294,959 households visited, and 1,027,763 household members were interviewed, with a response rate of 93.0%. Two types of structured questionnaires were used: individual and household. A detailed explanation of the survey methodology has been described in detail elsewhere [17]. In this analysis, we used information collected from 10,206 women with children under five years old in South Sulawesi Province and 2,436 women with children under five years old in West Sulawesi Province.

The 2018 Indonesia Basic Health Survey survey included households from all 33 provinces and 497 districts/cities of Indonesia. The 2018 survey used the same sampling method as the previous national survey. There were 30,000 census blocks visited, 30,000 census blocks targeted, and 300,000 households visited. In South Sulawesi, 13,811 households were visited out of 13,840 target households. Meanwhile, in West Sulawesi Province, 2,910 households visited

2,960 target households. Two types of structured questionnaires were used: individual and household [18]. In this analysis, we collected information from 10,862 women with children under five years old in South Sulawesi Province and 2,663 women with children under five years old in West Sulawesi Province.

South Sulawesi and West Sulawesi Province were chosen because of the high prevalence of stunting, and it was previously one province divided in 2007. The prevalence of stunting in West Sulawesi Province in 2018 was 41.59%, the second highest in Indonesia. At the same time, the prevalence of stunting in South Sulawesi was 35.74%, making it the fourth highest stunting in Indonesia [18].

## Outcome variable

The primary outcome of this study is stunting (low height for age) of the children under five years old measured at the time of the survey. Stunting was a nutritional status indicator based on height for age, or a child's height reached a certain age. Based on WHO growth standards, the height indicator for a period is determined based on the z-score or height deviation from average height. The WHO Anthro software was used to calculate the z-score. Stunted was categorized as 1 for stunted children and 0 for those not stunted. The limit for the nutritional status category according to the height index/age is [19, 20]:

- • Stunted: < -3.0 SD to -2.0 SD

- • Normal: ≥ -2.0 SD

## Potential predictors

15 potential predictors were analysed and categorized into three main categories, household characteristics, maternal characteristics, and children characteristics. These variables were selected based on the literature review on possible predictors of stunting in children under five years. Variables in the household characteristics were household wealth index, number of household members, and number of children under five in the household. The House wealth index was created to describe the household's economic status. We constructed this variable from the variables of the household facilities and assets using the Principal Component Analysis [21].

There were six variables in the maternal characteristic group, maternal education, maternal employment status, maternal age, maternal weight, maternal height, and maternal body mass index (BMI)s. In the children's characteristics group, there were six variables: sex, age, birthweight, breastfeeding status, diarrhea, and acute respiratory infection.

## Data analysis

Data were analysed using Stata version 11 (StataCorp, College Station, TX, USA). Weighting was carried out on all analyzes. Descriptive analysis was employed to examine the data, followed by bivariate analysis to determine the distribution by stunting status. The measure of association between the outcomes of interest (stunting) with the independent variables were the Prevalence Ratios (PRs) at their 95% condence intervals and p-values of <0.05 showed statistically signicant associations between the outcomes and the independent variables. The PRs were used instead of Odds Ratios (ORs) because the prevalence rates of stunting and underweight were more than 10%. The ORs tend to overestimate the strength of association in such scenarios. PRs at both the bivariate and multivariate analysis level was estimated using the

Poisson regression analysis model [22] After testing for co-linearity, covariates with p-value ≤0.2 at the bivariate analysis were considered for the multivariable model. Covariates with p-values < 0.05 after multivariable analysis were considered as determinants of stunting and underweight.

### Informed consent and ethics clearance

All the member of the households interviewed in the 2013 and 2018 Indonesia Basic Health Survey were explained about the detail of the survey and informed about the consent. Written informed consent was obtained from all participants for inclusion in the study. Informed consent statement was printed on the form that signed by the member of the households who agree to participate in the survey.

The ethical approval of this study was obtained from the Health Research Ethics Commission, Faculty of Medicine, Hasanuddin University, number 137/UN4.6.4.5.31/PP36/2022.

## Results

### Characteristics of study respondents

A total of 3641 and 4423 respondents of children under five years old in South Sulawesi from 2013 and 2018 Indonesia Basic Health Survey, respectively; and 804 and 1059 respondents from 2013 and 2018 Indonesia Basic Health Survey in West Sulawesi, respectively. The mean age of children in South Sulawesi and West Sulawesi Province was 31.1 months and 30.8 months, respectively, on 2013 survey compare to 29.6 months and 29.1 months on the 2018 survey. The distribution of household by household, maternal, and child characteristics for children under five years old are presented in Table 1. A quarter of respondents in South Sulawesi were in quantile 4 of the household wealth index, whereas in West Sulawesi, respondents are mostly in quantile 1. The number of household members in South and West Sulawesi was five to seven people per household in half of the respondents, with one child under five in almost three-quarters of respondents. The mother's education level was mainly primary school or did not attend school. In addition, most mothers were not working. More than half of the children under five were aged 24–59 months. Moreover, most children had a normal birth weight and were ever breastfed.

### Bivariate and multivariable analysis of determinants of stunting

Table 2 shows that without adjusting for any other covariates, there was a significant association between stunting and household wealth index, the number of children under five years in the household, maternal education, maternal height, maternal age, maternal BMI, child's age at the time of the interview, birthweight, ever breastfed and stunting based on 2013 Indonesia basic health survey in South Sulawesi. On the 2018 Indonesia Basic Health Survey, the number of children under five years in the household, maternal education, maternal height, maternal weight, child's age at the time of the interview, weight at birth, and ever breastfed had a significant association with stunting for poissin regression.

Table 3 shows a significant association between stunting and number of household member, child's age at the interview, and history of diarrhea on the 2013 Indonesia basic health survey in West Sulawesi without adjusting for any other covariates. For the 2018 Indonesia Basic Health Survey, maternal education, maternal weight, child's age at the timeof interview, history of acute respiratory infection (ARI), and ever breastfed had a significant association with stunting.

**Table 1. Frequency distribution of respondents based on several characteristics in the Provinces of South Sulawesi and West Sulawesi, 2013 and 2018 Indonesia Basic Health Survey.**

| Variable | South Sulawesi | | | | West Sulawesi | | | |
|---|---|---|---|---|---|---|---|---|
| | 2013 | | 2018 | | 2013 | | 2018 | |
| | n | (%) | n | (%) | n | (%) | n | (%) |
| **Household Characteristics** | | | | | | | | |
| Household wealth index | | | Data not available | | | | Data not available | |
| Quantile 5 (the richest) | 607 | (16.7) | | | 53 | (6.6) | | |
| Quantile 4 | 939 | (25.8) | | | 109 | (13.6) | | |
| Quantile 3 (intermediate) | 814 | (22.4) | | | 138 | (17.2) | | |
| Quantile 2 | 585 | (16.0) | | | 190 | (23.6) | | |
| Quantile 1 (poorest) | 696 | (19.1) | | | 314 | (39.1) | | |
| Number of household members | | | | | | | | |
| 1–4 | 1427 | (39.2) | 1438 | (32.5) | 313 | (38.9) | 410 | (38.7) |
| 5–7 | 1845 | (50.7) | 2353 | (53.2) | 425 | (52.9) | 520 | (49.1) |
| 8+ | 369 | (10.1) | 632 | (14.3) | 66 | (8.2) | 129 | (12.2) |
| Number of children under five in the household | | | | | | | | |
| 1 | 2600 | (71.4) | 3070 | (69.4) | 543 | (67.5) | 723 | (68.3) |
| 2 | 921 | (25.3) | 1166 | (26.4) | 243 | (30.2) | 296 | (28.0) |
| 3+ | 120 | (3.3) | 187 | (4.2) | 18 | (2.3) | 40 | (3.7) |
| **Mother Characteristics** | | | | | | | | |
| Mother's education | | | | | | | | |
| Academy/university | 369 | (10.1) | 706 | (15.9) | 63 | (7.8) | 130 | (12.3) |
| Senior high school | 859 | (23.6) | 1166 | (26.4) | 123 | (15.3) | 238 | (22.5) |
| Junior high school | 728 | (20.0) | 892 | (20.2) | 149 | (18.5) | 193 | (18.2) |
| Elementary/No school | 1685 | (46.3) | 1659 | (37.5) | 469 | (58.3) | 498 | (47.0) |
| Mother's employment | | | | | | | | |
| Doesn't work | 2746 | (75.4) | 2851 | (64.5) | 566 | (70.4) | 603 | (56.9) |
| Working | 875 | (24.1) | 1545 | (34.9) | 236 | (29.4) | 448 | (42.3) |
| School | 20 | (0.5) | 27 | (0.6) | 2 | (0.2) | 8 | (0.8) |
| Mother's height | | | | | | | | |
| >160cm | 437 | (10.0) | 513 | (11.6) | 58 | (7.2) | 79 | (7.4) |
| 151–160cm | 1892 | (51.9) | 2370 | (53.6) | 370 | (46.0) | 485 | (45.8) |
| 140 cm | 1312 | (36.1) | 1540 | (34.8) | 376 | (46.7) | 495 | (46.8) |
| **Child Characteristics** | | | | | | | | |
| Sex of child | | | | | | | | |
| Female | 1735 | (47.7) | 2126 | (48.1) | 378 | (47.0) | 485 | (45.8) |
| Male | 1906 | (52.3) | 2297 | (51.9) | 426 | (53.0) | 574 | (54.2) |
| Age | | | | | | | | |
| <12 months | 640 | (17.6) | 889 | (20.1) | 147 | (18.3) | 221 | (20.9) |
| 12–23 months | 692 | (19.0) | 865 | (19.6) | 138 | (17.2) | 213 | (20.1) |
| 24–59 months | 2309 | (63.4) | 2669 | (60.3) | 519 | (65.5) | 625 | (59.0) |
| Weight at birth | | | | | | | | |
| 2500 g | 941 | (25.8) | 1976 | (44.7) | 209 | (26.0) | 465 | (43.9) |
| < 2500 g | 54 | (1.5) | 163 | (3.7) | 8 | (1.0) | 26 | (2.5) |
| Don't know | 2646 | (72.7) | 2284 | (51.6) | 587 | (73.0) | 568 | (53.6) |
| Breastfeeding status | | | | | | | | |
| Yes | 1231 | (33.8) | 1607 | (36.3) | 271 | (33.7) | 400 | (37.8) |
| Not | 101 | (2.8) | 129 | (2.9) | 14 | (1.7) | 24 | (2.3) |
| Don't know | 2309 | (63.4) | 2687 | (60.8) | 519 | (64.6) | 635 | (59.9) |

**Table 2. Bivariate analysis of stunting determinants in children under five years in South Sulawesi, 2013 and 2018 Indonesia Basic Health Survey.**

| | 2013 | | | 2018 | | |
|---|---|---|---|---|---|---|
| | Stunting | Bivariate Analysis | | Stunting | Bivariate Analysis | |
| | % | CPR[b] (95% CI) | p | % | CPR[b] (95% CI) | p |
| **Household characteristics** | | | | | | |
| Household wealth index | | | | Data not available | | |
| Quantile 5 (the richest) | 29.3 | 1.0 | | | | |
| Quantile 4 | 33.2 | 1.1 (0.9–1.4) | 0.159 | | | |
| Quantile 3 (intermediate) | 35.1 | 1.1 (0.9–1.4) | 0.212 | | | |
| Quantile 2 | 41.5 | 1.4 (1.1–1.7) | 0.001 | | | |
| Quantile 1 (poorest) | 41.7 | 1.4 (1.1–1.7) | 0.002 | | | |
| Number of household members | | | | | | |
| 1–4 | 34.8 | 1.0 | | 32.0 | 1.00 | |
| 5–7 | 35.9 | 1.0 (0.9–1.2) | 0.455 | 30.1 | 0.9 (0.8–1.1) | 0.359 |
| 8+ | 40.9 | 1.1 (0.9–1.4) | 0.217 | 27.8 | 0.9 (0.7–1.1) | 0.152 |
| Number of children under five in the household | | | | | | |
| 1 | 35.2 | 1.00 | | 31.8 | 1.00 | |
| 2 | 38.5 | 1.2 (1.1–1.3) | 0.003 | 27.3 | 0.9 (0.7–0.9) | 0.038 |
| 3+ | 32.5 | 0.9 (0.7–1.4) | 0.936 | 27.1 | 0.9 (0.6–1.3) | 0.413 |
| **Mother Characteristics** | | | | | | |
| Mother's education | | | | | | |
| Academy/university | 29.3 | 1.0 | | 22.5 | 1.00 | |
| Senior high school | 33.2 | 1.0 (0.8–1.3) | 0.936 | 26.3 | 1.2 (0.9–1.5) | 0.174 |
| Junior high school | 37.8 | 1.2 (0.9–1.6) | 0.120 | 31.0 | 1.4 (1.1–1.7) | 0.004 |
| Elementary/No school | 38.0 | 1.3 (1.1–1.6) | 0.028 | 37.4 | 1.7 (1.4–2.0) | <0.001 |
| Mother's employment | | | | | | |
| Unemployed | 35.9 | 1.0 | | 31.2 | 1.00 | |
| Employed | 36.0 | 0.9 (0.8–1.1) | 0.441 | 28.8 | 0.9 (0.8–1.1) | 0.228 |
| Currently at school | 30.0 | 0.4 (0.1–1.2) | 0.093 | 20.6 | 0.7 (0.3–1.6) | 0.371 |
| Mother's height | | | | | | |
| >160cm | 31.0 | 1.0 | | 23.9 | 1.00 | |
| 151–160cm | 31.4 | 1.0 (0.8–1.2) | 0.916 | 25.9 | 1.1 (0.9–1.4) | 0.519 |
| ≤150 cm | 43.6 | 1.4 (1.2–1.7) | 0.001 | 40.1 | 1.7 (1.3–2.1) | <0.001 |
| Mothers's age[a] | - | 1.0 (0.9–1.0) | 0.861 | - | 1.00 (0.9–1.0) | 0.990 |
| Mother's weight[a] | - | 0.9 (0.97–0.98) | <0.001 | - | 0.9 (0.98–0.99) | <0.001 |
| Body mass index (BMI)[a] | - | 0.9 (0.96–0.99) | 0.002 | - | 0.9 (0.9–1.0) | 0.339 |
| **Child Characteristics** | | | | | | |
| Sex of child | | | | | | |
| Female | 34.9 | 1.0 | | 30.2 | 1.00 | |
| Male | 36.9 | 1.1 (0.9–1.2) | 0.082 | 32.3 | 1.1 (0.9–1.2) | 0.166 |
| Age | | | | | | |
| <12 months | 20.6 | 1.0 | | 15.9 | 1.0 | |
| 12–23 months | 38.3 | 2.3 (1.8–2.8) | <0.001 | 32.4 | 2.2 (1.7–2.8) | <0.001 |
| 24–59 months | 39.5 | 2.3 (1.8–2.8) | <0.001 | 36.1 | 2.5 (2.0–3.1) | <0.001 |
| Weight at birth | | | | | | |
| 2500 g | 29.0 | 1.0 | | 28.8 | 1.00 | |
| < 2500 g | 55.6 | 2.3 (1.7–3.0) | <0.001 | 45.4 | 1.7 (1.3–2.1) | <0.001 |

*(Continued)*

**Table 2.** (Continued)

| | 2013 | | | 2018 | | |
| | Stunting | Bivariate Analysis | | Stunting | Bivariate Analysis | |
| | % | CPR[b] (95% CI) | p | % | CPR[b] (95% CI) | p |
|---|---|---|---|---|---|---|
| Breastfeeding status | | | | | | <0.001 |
| Yes | 28.8 | 1.0 | 0.006 | 23.8 | 1.00 | |
| Not | 42.6 | 1.5 (1.1–2.0) | | 25.6 | 1.5 (1.3–1.7) | |
| Sufferedd from diarrhea | | | | | | |
| No | 35.6 | 1.0 | | 31.0 | 1.00 | |
| Yes. 2 weeks ago | 36.7 | 1.0 (0.8–1.3) | 0.861 | 33.5 | 1.2 (0.9–1.5) | 0.108 |
| Yes. > 2 weeks ago | 42.1 | 1.1 (0.9–1.5) | 0.281 | 36.6 | 1.2 (0.9–1.6) | 0.134 |
| Suffered from ARI[c] in the last month | | | | | | |
| Not | 35.2 | 1.0 | | 31.3 | 1.00 | |
| Yes | 38.3 | 1.1 (0.9–1.3) | 0.078 | 32.0 | 1.1 (0.8–1.5) | 0.635 |

[a]continuous variable

[b]CPR = Crude Prevalence Ratio

[c]ARI = Acute Respiratory Infection

The results of the multivariate analysis are presented in Table 4. On the 2013 Indonesia Basic Health Survey in South Sulawesi Province, children under five years who lived in a household with two children under five had a prevalence of stunting 1.24 times greater than children in a household with only one child under five years (APR:1.24; 95% CI: 1.1–1.3; p-value = 0.006). Based on the maternal characteristics, the prevalence of stunting increased with the increase of maternal BMI by one point (APR: 1.1; 95% CI: 1.03–1.1; p-value = <0.001). Conversely, the prevalence of stunting would be expected to decrease by a factor of 0.9 if maternal weight increases by one point (APR: 0.9; 95% CI: 0.94–0.97; p-value = <0.001).

The prevalence of stunting in children aged 12–23 months was about two times higher than in children aged less than 12 months (APR: 2.2; 95% CI: 1.7–2.7; p-value = <0.001). Similarly, children aged 24–59 months had a prevalence of stunting about two times that among children younger than 12 months (APR: 2.2; 95% CI: 1.8–2.7; p-value = <0.001). The weight at birth was also associated with stunted children under five years. Children under five years whose birthweight was less than 2500 grams had about two times higher stunting prevalence than those whose birthweight was 2500 grams or higher (APR: 1.9; 95% CI: 1.5–2.5; p-value = <0.001).

Meanwhile, Table 4 shows five predictors as the determinant of stunting in children under five in the 2018 survey. The prevalence of stunting would be expected to decrease by a factor of 0.9 if maternal weight increased by one point (APR: 0.9; 95% CI: 0.98–0.99; p-value = 0.005). Mothers with no education or with education in primary school had a prevalence of stunting in their children under five years, 1.5 times higher than mothers with academy/university education (APR: 1.5; 95% CI: 1.3–1.9; p-value = <0.001). Meanwhile, the prevalence of stunting in mothers with education in middle school was 1.3 times higher than in mothers with academy/university education (APR: 1.3; 95% CI: 1.1–1.6; p-value = 0.014). The prevalence of stunting in children under five years in mothers with a height less than 151 cm was 2.2 times higher than in mothers with more than 160 cm (APR: 1.3; 95% CI: 1.5–3.1; p-value = <0.001). Meanwhile, mothers with a height of 151–160 cm had a 1.5 times higher prevalence of stunting in children under five years than mothers with more than 160 cm (APR: 1.3; 95% CI: 1.1–1.6; p-value = 0.014).

**Table 3. Bivariate analysis of stunting determinants in children's age under five years in West Sulawesi, 2013 and 2018 Indonesia Basic Health Survey.**

| | 2013 | | | 2018 | | |
|---|---|---|---|---|---|---|
| | Stunting | Bivariate Analysis | | Stunting | Bivariate Analysis | |
| | % | APR[b] (95% CI) | p | % | APR[b] (95% CI) | p |
| **Household characteristics** | | | | | | |
| Household wealth index | | | | Data not available | | |
| Quantile 5 (the richest) | 24.5 | 1.0 | | | | |
| Quantile 4 | 28.4 | 1.0 (0.5–2.1) | 0 933 | | | |
| Quantile 3 (intermediate) | 36.2 | 1.3 (0.7–2.6) | 0 369 | | | |
| Quantile 2 | 48.4 | 1.9 (1.0–3.6) | 0.063 | | | |
| Quantile 1 (poorest) | 44.6 | 1.6 (0.8–3.0) | 0.148 | | | |
| Number of household members | | | | | | |
| 1–4 | 42.2 | 1.0 | | 39.5 | 1.0 | 0.347 |
| 5–7 | 36.9 | 0.9 (0.7–1.2) | 0.553 | 36.1 | 0.9 (0.8–1.1) | 0.346 |
| 8+ | 56.1 | 1.5 (1.2–1.8) | 0.001 | 36.3 | 0.9 (0.6–1.3) | 0.634 |
| Number of children under five in the household | | | | | | |
| 1 | 39.6 | 1.0 | | 37.3 | 1.0 | |
| 2 | 41.2 | 1.1 (0.9–1.4) | 0.340 | 39.6 | 1.1 (0.9–1.3) | 0.574 |
| 3+ | 61.1 | 1.7 (1.1–2.5) | 0.018 | 29.2 | 0.8 (0.4–1.5) | 0.466 |
| **Mother Characteristics** | | | | | | |
| Mother's education | | | | | | |
| Academy/university | 34.9 | 1.00 | | 20.7 | 1.0 | |
| Senior high school | 30.9 | 0.7 (0.4–1.3) | 0.305 | 37.2 | 1.8 (1.2–2.7) | 0.004 |
| Junior high school | 40.3 | 1.2 (0.7–2.0) | 0.485 | 40.5 | 2.0 (1.3–2.9) | 0.001 |
| Elementary/No school | 43.9 | 1.3 (0.8–2.1) | 0.305 | 42.1 | 2.0 (1.4–2.9) | < 0.001 _ |
| Mother's employment | | | | | | |
| Unemployed | 39.1 | 1.0 | | 37.7 | 1.0 | |
| Employed | 43.6 | 1.1 (0.9–1.3) | 0.489 | 37.6 | 0.9 (0.8–1.2) | 0.978 |
| Currently at school | 50.0 | 0.9 (0.2–5.3) | 0.925 | 35.7 | 0.9 (0.3–2.7) | 0.923 |
| Mother's height | | | | | | |
| >160cm | 35.4 | 1.0 | | 36.0 | 1.0 | |
| 151–160cm | 34.9 | 1.0 (0.7–1.5) | 0.946 | 30.1 | 0.8 (0.6–1.2) | 0.328 |
| 150 cm | 49.9 | 1.4 (0.9–2.1) | 0.085 | 45.5 | 1.3 (0.9–1.8) | 0.214 |
| Mother's age[a] | - | 1.0 (0.9–1.0) | 0.939 | - | 0.9 (0.9–1.0) | 0.183 |
| Mother's weight[a] | - | 1.0 (0.9–1.0) | 0.057 | - | 0.9 (0.97–0.99) | 0.007 |
| Body mass index (BMI)[a] | - | 1.0 (0.9–1.0) | 0.322 | - | 0.9 (0.95–1.0) | 0.167 |
| **Child characteristics** | | | | | | |
| Sex of child | | | | | | |
| Female | 38.6 | 1.0 | | 34.9 | 1.0 | |
| Male | 42.3 | 1.1 (0.9–1.4) | 0.274 | 42.5 | 1.2 (0.9–1.4) | 0.063 |
| Age | | | | | | |
| <12 months | 23.1 | 1.0 | | 19.0 | 1.0 | |
| 12–23 months | 41.3 | 1.8 (1.2–2.7) | 0.006 | 40.4 | 1.9 (1.3–2.7) | 0.001 |
| 24–59 months | 45.3 | 1.9 (1.3–2.7) | 0.001 | 45.6 | 2.3 (1.7–3.1) | < 0.001 |
| Weight at birth | | | | | | |
| 2500 g | 35.4 | 1.0 | | 35.5 | 1.0 | |
| < 2500 g | 12.5 | 0.3 (0.0–2.5) | 0.282 | 50.0 | 1.4 (0.8–2.3) | 0.210 |
| Breastfeeding status | | | | | | |
| Yes | 31.4 | 1.0 | | 30.0 | 1.0 | |

*(Continued)*

**Table 3.** (Continued)

| | 2013 | | | 2018 | | |
|---|---|---|---|---|---|---|
| | Stunting | Bivariate Analysis | | Stunting | Bivariate Analysis | |
| | % | APR[b] (95% CI) | p | % | APR[b] (95% CI) | p |
| Not | 42.9 | 1.1 (0.4–2.7) | 0.840 | 25.0 | 1.5 (1.2–1.8) | <0.001 |
| Suffered from diarrhea | | | | | | |
| No | 40.3 | 1.0 | | 38.7 | 1.0 | |
| Yes. 2 weeks ago | 33.3 | 0.8 (0.5–1.4) | 0.474 | 41.3 | 1.0 (0.7–1.4) | 0.949 |
| Yes. > 2 weeks ago | 52.8 | 1.4 (1.03–2.0) | 0.033 | 40.7 | 1.1 (0.8–1.6) | 0.565 |
| Suffered from ARI[c] in the last month | | | | | | |
| Not | 39.5 | 1.0 | | 38.6 | 1.0 | |
| Yes | 46.2 | 1.1 (0.9–1.4) | 0.398 | 56.0 | 1.5 (1.04–2.3) | 0.031 |

[a]continuous variable

[b]CPR = Crude Prevalence Ratio

[c]ARI = Acute Respiratory Infection

The prevalence of stunting in children aged 12–23 months was about two times higher than in children aged less than 12 months (APR: 2.2; 95% CI: 1.7–2.8; p-value = <0.001). Similarly, children aged 24–59 months had a prevalence of stunting about 2.5 times higher than among children younger than 12 months (APR: 2.5; 95% CI: 2.0–3.2; p-value = <0.001). The weight at birth was also associated with stunted children under five years. Children under five years whose birthweight was less than 2500 grams had about 1.5 times higher stunting prevalence than those whose birthweight was 2500 grams or higher (APR: 1.5; 95% CI: 1.2–1.9; p-value = <0.001).

Table 4 shows that in West Sulawesi Province, three predictors were the determinants of stunting in children under five years based on the 2013 Indonesia Basic Health Survey. The prevalence of stunting in children under five years living in poor households was 1.9 times higher than in those living in the richest households (APR: 1.9; 95% CI: 1.1–3.3; p-value = 0.021). Meanwhile, children under five who lived in a household with three or more children under five had a prevalence of stunting 1.8 times greater than children with only one child under five years (APR:1.8; 95% CI: 1.2–2.7; p-value = 0.002). Moreover, the prevalence of stunting in children aged 12–23 months was 1.8 times higher than in children aged less than 12 months (APR: 1.8; 95% CI: 1.2–2.6; p-value = 0.006). Similarly, children aged 24–59 months had a prevalence of stunting about 1.9 times higher than among children younger than 12 months (APR: 1.9; 95% CI: 1.3–2.7; p-value = 0.001).

On the 2018 Indonesia Basic Health Survey in West Sulawesi Province, six predictors were the determinants of stunting in children under five. The prevalence of stunting increased with the increase of maternal BMI by one point (APR: 1.02; 95% CI: 1.01–1.03; p-value = 0.004). Conversely, the prevalence of stunting would be expected to decrease by a factor of 0.9 if maternal weight increases by one point (APR: 0.9; 95% CI: 0.95–0.98; p-value = <0.001). Mothers with no education or with education in primary school had a prevalence of stunting in their children under five years, 1.9 times higher than mothers with academy/university education (APR: 1.9; 95% CI: 1.3–2.7; p-value = 0.001). Similarly, the prevalence of stunting in mothers with education in middle school was 1.9 times higher than in mothers with academy/university education (APR: 1.9; 95% CI: 1.3–2.7; p-value = 0.001). Moreover, mothers with education in high school had a prevalence of stunting in their children under five years, 1.8

**Table 4. Determinants of stunting in children under five (toddlers) in South and West Sulawesi Provinces, 2013 and 2018 Indonesia Basic Health Survey.**

| | 2013 | | | | 2018 | | | |
| --- | --- | --- | --- | --- | --- | --- | --- | --- |
| | South Sulawesi | | West Sulawesi | | South Sulawesi | | West Sulawesi | |
| | Multivariate Analysis | | Multivariate Analysis | | Multivariate Analysis | | Multivariate Analysis | |
| | PR (95% CI) | p | PR (95% CI) | p | PR (95% CI) | p | PR (95% CI) | p |
| **Household Characteristics** | | | | | | | | |
| Household Wealth Index | | | | | Data not avalaible | | | |
| Quantile 5 (the richest) | - | - | 1.0 | | | | | |
| Quantile 4 | - | - | 1.0 (0.5–2.0) | 0.916 | | | | |
| Quantile 3 (middle) | - | - | 1.4 (0.8–2.4) | 0.248 | | | | |
| Quantile 2 | - | - | 1.9 (1.1–3.3) | 0.021 | | | | |
| Quantile 1 (poorest) | - | - | 1.6 (0.9–2.7) | 0.097 | | | | |
| Number of household members | | | | | | | | |
| 1–4 | - | - | - | - | - | - | - | - |
| 5–7 | - | - | - | - | - | - | - | - |
| 8+ | - | - | - | - | - | - | - | - |
| Number of children under five in the household | | | | | | | | |
| 1 | 1.0 | | 1.0 | | - | - | - | - |
| 2 | 1.2 (1.1–1.3) | 0.006 | 1.2 (0.9–1.4) | 0.178 | - | - | - | - |
| 3+ | 1.0 (0.7–1.5) | 0.968 | 1.8 (1.2–2.7) | 0.0002 | - | - | - | - |
| **Mother Characteristics** | | | | | | | | |
| Mother's height | | | | | | | | |
| >160cm | - | - | - | - | 1.0 | | | |
| 151–160cm | - | - | - | - | 1.5 (1.03–2.2) | 0.032 | - | - |
| 150 cm | - | - | - | - | 2.2 (1.5–3.1) | <0.001 | - | - |
| Mother's education | | | | | | | | |
| Academy/university | - | - | - | - | 1.0 | | 1.0 | |
| High school | - | - | - | - | 1.1 (0.9–1.4) | 0.338 | 1.8 (1.2–2.6) | 0.004 |
| Middle school | - | - | - | - | 1.3 (1.1–1.6) | 0.014 | 1.9 (1.3–2.7) | 0.001 |
| Primary school/No school | - | - | - | - | 1.5 (1.3–1.9) | <0.001 | 1.9 (1.3–2.7) | 0.001 |
| Mother's weight | 0.9 (0.94–0.97) | <0.001 | - | - | 0.9 (0.98–0.99) | 0.005 | 0.9 (0.95–0.98) | <0.001 |
| Body mass index (BMI) | 1.1 (1.03–1.1) | <0.001 | - | - | - | - | 1.02 (1.01–1.4) | 0.004 |
| **Child Characteristics** | | | | | | | | |
| Sex of baby | | | | | | | | |
| Female | - | - | - | - | - | - | 1.0 | |
| Male | - | - | - | - | - | - | 1.2 (1.1–1.4) | 0.035 |
| Age | | | | | | | | |
| <12 months | 1.0 | | 1.0 | | 1.0 | | 1.0 | |
| 12–23 months | 2.2 (1.7–2.7) | <0.00 | 1.8 (1.2–2.6) | 0.006 | 2.2 (1.7–2.8) | <0.001 | 2.1 (1.4–3.0) | <0.001 |
| 24–59 months | 2.2 (1.8–2.7) | 1 <0.001 | 1.9 (1.3–2.7) | 0.001 | 2.5 (2.0–3.2) | <0.001 | 2.6 (1.9–3.6) | <0.001 |
| Weight at birth | | | | | | | | |
| 2500 g | 1.0 | | - | - | 1.0 | | - | - |
| < 2500 g | 1.9 (1.5–2.5) | <0.001 | - | - | 1.5 (1.2–1.9) | <0.001 | - | - |
| Suffered from ARI in the last month | | | | | | | | |
| Not | - | - | - | - | - | - | 1.0 | |
| Yes | - | - | - | - | - | - | 1.6 (1.04–2.5) | 0.030 |

times higher than mothers with academy/university education (APR: 1.8; 95% CI: 1.2–2.6; p-value = 0.004).

The prevalence of stunting in children aged 12–23 months was about two times higher than in children aged less than 12 months (APR: 2.1; 95% CI: 1.4–3.0; p-value = <0.001). Similarly, children aged 24–59 months had a prevalence of stunting about 2.6 times higher than among children younger than 12 months (APR: 2.6; 95% CI: 1.9–3.6; p-value = <0.001). Male compared to female had 1.2 times higher prevalence of stunting in children under five (APR: 1.2; 95% CI: 1.01–1.4; p-value = 0.035). The prevalence of stunting among children who suffered from acute respiratory infection (ARI) in the last month proceeding the survey was about 1.6 times higher than among children who did not suffer from AsRI in the last month proceeding the survey (APR: 1.6; 95% CI: 1.04–2.5; p-value = 0.030).

## Discussion

Our study found that the determinants of stunting among children under five years old in South Sulawesi based on the 2013 and 2018 Indonesia Basic Health Survey was household wealth index, the number of the household member under five years, maternal height, maternal weight, maternal education, maternal body mass index, child's age, and weight at birth. Whereas in West Sulawesi, was the household wealth index, number of household members under five years, maternal education, maternal weight, maternal body mass index, sex of the children, child's age, and history of acute respiratory infection.

At the household level, the household index was associated with stunting in South Sulawesi and West Sulawesi. The index reflects a household's ability to purchase goods, good quality foods, and adequate health services [23]. This finding shows that children's health depends on the household's socioeconomic status. Studies have shown that a low household wealth index is a key predictor factor for stunting among children under five years of age [9] and is closely related to stunting through food insecurity status in the household [24–26] and the fulfillment of children's minimum dietary needs [24, 26, 27]. This finding is similar to the other studies from Bangladesh [11], India [28], Nepal [9], Maldives, Pakistan [12], and Indonesia [28] that children aged 0–59 months from poorer households were at higher odds of being stunted than wealthy households. Richer households have better power in purchasing food and other consumer goods needed for children's health [9]. However, children from poorer households have limited food access, making them more vulnerable to growth failures [12, 29].

Children in low-income groups and those living in low-income neighborhoods are at higher risk of stunting. Poverty and low socioeconomic status have a more detrimental effect on linear growth than body weight. Economic inequality is an independent determinant for malnourished children. Research has shown that poor children are at higher risk of malnutrition and are more likely to experience growth retardation. Countries with greater levels of economic inequality tend to have poorer health status than countries with more economic equality. Developing countries remain vulnerable to food insecurity, poor access to health services, malnutrition, and increased morbidity and mortality, and economic growth's health and nutritional benefits tend to be concentrated in the economically advantaged groups [30].

Family size and the number of household members under five years were other factors related to stunting. These findings are consistent with studies conducted in Brazil and South Africa, where children living in households with many members are more likely to develop stunting. A large household may cause resource depletion, reduced availability, and competition for available food. Family size might cause the improper allocation of food, which can lead to children's poor health and not optimum nutritional status. It is also in line with studies conducted in Southern Brazil [31], South Africa [32], and Ethiopia [10, 23].

Our study shows that maternal height was related to stunting in the two provinces, South Sulawesi and West Sulawesi. Previous studies have shown that children of short stature mothers (<150 cm) were more likely to be stunted [33]. Studies from five South Asian Countries and 35 low and middle-income regions (LMCIs) showed the association of shorter maternal height with stunting among children aged 12–59 months. A mother's height has been used to assess the intergenerational health linkage between a mother and her offspring. This could be due to genetic and environmental factors, such as nutritional intake, diet, and culture, that influence mothers during childhood and later the growth of their children [12, 28, 33, 34]. In addition to maternal height, maternal education also influenced the incidence of stunting in children. This result is in line with studies conducted in Pakistan, Bangladesh, Kenya, and Indonesia. Children whose mothers have never attended formal education are more likely to be stunted than children whose mothers have formal education. Similarly, a study conducted in Nigeria, Pakistan, and Pune revealed that low maternal education is a risk factor for children's stunting [35].

Low birth weight shows a strong relationship with stunting in children under five. This finding is supported by research conducted in Pakistan and Mexico which found that children under 24 months of age who were born with low birth weight were three times more likely to be stunted than children in the same age group with normal birth weight [36, 37]. In areas with high low birth weight cases, prevention of intrauterine growth retardation, premature birth, and maternal malnutrition should be the basis of public health intervention strategies to prevent infant stunting [9]. Research using the 2010 Basic Health Survey found that low birth weight was associated with infant body stunting in Indonesia. Low birth weight is a predisposing factor for achieving growth after birth. Evidence suggests that poor early growth retardation coincides with suboptimal cognitive development, and stunted growth of internal organs can result in low cognitive abilities and an increased risk of chronic disease later in life. A study in Zimbabwe found that growth in low birth weight infants was far behind that of normal weight infants, and a significant difference in length was seen at 12 months of age [38]. The association between low birth weight and childhood malnutrition may also be related to the increased vulnerability of children with low birth weight to infections and lower respiratory infections and the increased risk of complications, including sleep apnea, jaundice, anemia, chronic lung disorders, fatigue, and loss of appetite, compared with children with normal birth weights [39].

In addition, this study also found that the sex of the child influenced stunting. Various studies show that malnutrition occurs in both sexes differently. Stunting is more common in boys than girls. It is in agreement with other studies among school-age children in Ethiopia. India, and Sir Lanka. This is because men's growth and development are more affected by environmental and nutritional stressors (including common childhood illnesses) than women and thus makes men more likely to be affected by chronic malnutrition [40]. Boys are more prone to malnutrition because they need more calories for growth and development [12].

The study also found that children over 12 months were significantly more likely to be stunted than children under 12 months. Another study also reported that the difference in length between low birth weight and normal weight infants increased with age from 12 months until the child reached two years of age [23]. The suboptimal growth associated with increasing age may stem from challenges transitioning from breastfeeding to complementary feeding [41]. Problems with child growth will occur if breastfeeding is not accompanied by adequate complementary feeding at the appropriate age. With increasing nutritional needs, if a child receives inadequate complementary feeding, linear growth retardation may occur [14, 23, 42]. Also, increased exposure to childhood diseases and conditions due to increasing age, such as poor food hygiene and environmental sanitation, can contribute to poor growth [23, 41].

Child growth is a complex process involving several nutritional and environmental factors [43].

In this study, several maternal factors were associated with child malnutrition. These findings suggest that nutrition intervention programs should include maternal sociodemographic factors to improve children's nutritional status. In addition, there is a need for strategies that focus on improving the nutritional status of mothers (through adequate food and micronutrient supplementation) and better care for infants and toddlers [12].

## Strength and limitations

The strength of this study is its large sample size that allows analysis of the relationship between variables and stunting in children under five years old. Moreover, the sampling weight in our analysis could reduce bias. The limitation of our study was the selection of independent variables depended on the variable available. The variable household wealth index could not be constructed because the variable of family facilities and assets were not available in the 2018 Indonesia Basic Health Survey.

## Conclusion

This study found that the determinants of stunting among children under five years old in South Sulawesi based on the 2013 and 2018 Indonesia Basic Health Survey was the number of household member under five years, maternal height, maternals education, maternal weight, maternal BMI, child's age, and weight at birth. Whereas in West Sulawesi, was the household wealth index, number of household members under five years, maternal education, maternal weight, maternal BMI, sex of the children, and child's age, history of acute respiratory infection.

Several maternal factors were associated with child malnutrition. These findings suggest that nutrition intervention programs should include maternal sociodemographic factors to improve children's nutritional status. Moreover, there is a need for strategies that focus on improving the nutritional status of mothers (through adequate food and micronutrient supplementation) and better care for infants and toddlers. Additionally, determinant factors from the household and children's level need to be addressed.

## Author Contributions

**Conceptualization:** Hayani Anastasia, Veni Hadju, Rudy Hartono, Samarang, Manjilala, Sirajuddin, Abdul Salam.

**Data curation:** Hayani Anastasia.

**Formal analysis:** Hayani Anastasia.

**Funding acquisition:** Hayani Anastasia, Veni Hadju, Rudy Hartono, Samarang, Manjilala, Sirajuddin, Abdul Salam.

**Investigation:** Hayani Anastasia.

**Methodology:** Hayani Anastasia, Veni Hadju, Rudy Hartono, Samarang, Manjilala, Sirajuddin, Abdul Salam.

**Supervision:** Atmarita.

**Validation:** Hayani Anastasia.

**Writing – original draft:** Hayani Anastasia.

**Writing – review & editing:** Hayani Anastasia, Veni Hadju, Rudy Hartono, Samarang, Manjilala, Sirajuddin, Abdul Salam, Atmarita.

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
