## [Decision Letter · Decision Letter 0]

16 Nov 2022

PONE-D-22-23326Determinants of stunting in children under five years old in South Sulawesi and West Sulawesi Province: 2013 and 2018 Indonesian Basic Health SurveyPLOS ONE

Dear Dr. Anastasia,

Thank you for submitting your manuscript to PLOS ONE. After careful consideration, we feel that it has merit but does not fully meet PLOS ONE’s publication criteria as it currently stands. Therefore, we invite you to submit a revised version of the manuscript that addresses the points raised during the review process.

We look forward to receiving your revised manuscript.

Kind regards,

Marly A. Cardoso, Ph.D.

Academic Editor

PLOS ONE

Journal Requirements

2. Please amend the manuscript submission data (via Edit Submission) to include author Atmarita.

3. We note that you have included the phrase “data not available” in your manuscript. Unfortunately, this does not meet our data sharing requirements. PLOS does not permit references to inaccessible data. We require that authors provide all relevant data within the paper, Supporting Information files, or in an acceptable, public repository. Please add a citation to support this phrase or upload the data that corresponds with these findings to a stable repository (such as Figshare or Dryad) and provide and URLs, DOIs, or accession numbers that may be used to access these data. Or, if the data are not a core part of the research being presented in your study, we ask that you remove the phrase that refers to these data.

Additional Editor Comments:

The two reviewers have pointed out important different issues for revision. In addition to the reviewer #1 comments on providing more methods details in the main text and prevalence with associated results in the abstract, I recomend to replace the statistical analyis using Prevalence Rates (it´s more apropriate for a cross-sectional study design with outcomes > 10% - please see the article by Barros & Hirakata. Alternatives for logistic regression in cross-sectional studies: an empirical comparison of models that directly estimate the prevalence ratio. https://bmcmedresmethodol.biomedcentral.com/articles/10.1186/1471-2288-3-21).

Reviewers' comments:

Reviewer's Responses to Questions

**Comments to the Author**

1. Is the manuscript technically sound, and do the data support the conclusions?

Reviewer #1: Yes

Reviewer #2: Partly

2. Has the statistical analysis been performed appropriately and rigorously? 

Reviewer #1: Yes

Reviewer #2: I Don't Know

3. Have the authors made all data underlying the findings in their manuscript fully available?

Reviewer #1: Yes

Reviewer #2: Yes

4. Is the manuscript presented in an intelligible fashion and written in standard English?

Reviewer #1: Yes

Reviewer #2: Yes

5. Review Comments to the Author

Reviewer #1: The authors studied the determinants of stunting among children aged under 5 years in the two different provinces in Indonesia.

Abstract

Are the areas districts, or provinces?

The study did not "try" but the study "determined"

The title mentions 2013 and 2018, but abstract has 2007. Despite these two provinces having been one province and only divided in 2007, be clear about the years you considered for the data analyzed in this study. You are saying "This study used 2007, 2013, and 2018 Indonesian Basic Health Surveys"????

Nowhere do you mention that the study was a retrospective/secondary data analysis? Was it not?

No sample size is mentioned.

In the results, mention the mean age of children. This is important. The study is about the under 5 year olds.

Also, I am uncomfortable that the abstract has not statistical results, just a mention of the factors.

The conclusion is not specific - needs to be revised. Since you have mentioned the households, maternal and child factors in the results, it is not necessary to repeat the sentence in a general manner like this in the conclusion. Linking the determinants with the recommendation is enough.

Introduction

Should consider other countries since Plos One is an international platform with a broader readership. Mentioning global, and LICMs in general is good in paragraph 1. At least in the 3rd paragraph starting with "Studies reveal various household, parental, and child factors are associated with stunting"., briefly elaborate on their place of origin.

Methods

Too summarized

It would interest the readers to read a brief description about the survey. Data came from a National survey, right? Does Indonesia has two provinces only - that description of Indonesia as a setting would be fair, until the readers get to understand that the two provinces; South Sulawesi and West Sulawesi were chosen based on what? Unless if they are the only provinces in Indonesia. Are they?

Much as you used previous data, it is still important to mention important aspects of the methods such as, sampling techniques used, sample size used in the national study. Also, what informed the extraction of which data to use in your study, especially on households, and maternal, for an example maternal age is missing in the variables you used to study the determinants; that's one important determinant but it is not mentioned anywhere. Again, why use maternal height, and leave weight? why not use maternal BMI? I am asking these questions to show you that you did not explain to us what informed the variables you have chosen? I acknowledge that it is not always easy to determine if there is a bias in a statistics study; it is quite difficult to conduct a zero bias study. But there is what we call BIAS DUE TO OMITTED VARIABLES (limitation). Except having to explain what informed selection of the variables you used, supposed some important variables like maternal age, weight, BMI were not there in the database, then, mention that as a limitation.

Elaborating on the measurements and tools that yielded HAZ, as well as how were the characteristics of children and mother collected during 2013, and 2018 would improve on a paper.

Results

Well presented.

It is always good to differentiate the OR for unadjusted and adjusted models with COR (crude odds ratio) and AOR (adjusted odd ratio) respectively - subject to the authors.

Discussion

I see you mention other countries in discussing and comparing the results in paragraph 2, such as Bangladesh, India, Nepal, Maldives, Pakistan - that's the standard, good. I also see Northern Brazil, South Africa and Ethiopia in paragraph 3 with only two references; #15 (Titaley et al, 2019 - Indonesia, 2013 data) and #25 (Fikadu et a, 2014 - South Ethiopian study). Please revise these references and ensure that correct references reflective of the findings are added.

Strengths and limitations

Is there any limitation in using a retrospective data analysis, especially for data collected in 2007, or 2013 for an example? 2018 is still within 5 years at least. bias on variables analyzed?

I believe that if the authors can objectively address the suggested correction above, then the paper would be excellent. The phenomenon studied is important, and stunting continues to wreck lives of children, especially in LMICs.

Reviewer #2: This cross sectional study investigates the determinants of stunting among children under 5 years old in two provinces in Indonesia: South Sulawesi and West Sulawesi. Using data from the 2013 and 2018 Indonesian Basic Health Survey, the authors conducted a multivariable logistic regression in order to find the determinants of stunting in each one of the two provinces. The findings show that the predictors of stunting include household characteristics, maternal characteristics, and children characteristics. The high prevalence of stunting in Indonesia and its underperformance about this problem over time highlights the importance of addressing the stunting issue in the region.

Overall, the manuscript is well written, leading to no ambiguity or doubts. However, there are some sections, specially those dedicated to the results and the discussion, which textual fluidity is poor. There is a great amount of information but the sentences are a bit disconnected, without interpretation, demanding extra attention from the reader to understand the main message.

In addition, the aim of the study was “to compare the determinants of stunting among children under five years old in South and West Sulawesi as these two provinces were one province and only divided in 2007”. However, what was done in the text was listing the determinants in each area. There was no comparison between the results from each area and, as a consequence, the comparison itself was not subject of the discussion nor was mentioned in the conclusion.

As the hypothesis and the justification are not clear in the manuscript, the connection between the aim and the results presented may be confusing.

Some additional issues could be identified over the paper.

- There are initials with no corresponding meanings (i.e. SSGI, IDD, e-PPGBM, ARI);

- There are some important methods description missing in the text. For example, there is no explanation about how variables were collected (interviews, anthropometric evaluation etc). Also, you do not describe which growth standards were used as reference to generate the z scores (World Health Organization?).

- Regarding the data analysis section, even though you mentioned Hosmer and Lemeshow suggestion, the paper would really benefit from some more details of the procedures to select the covariates and the level of significance considered.

- Tables: the independent variables presented in the tables are a little bit confusing. For example, does the “mother’s employment” question refer to paid jobs? “Working” may include informal, formal and even domestic work. Does the “School” category mean that the mother is at school (i.e. student) or that she works in a school? Other details should be reviewed (mother’s height categories; questions to children under 5 years old such as “Have you ever been breastfed?”).

- The information contained in the figures 3 and 4 do not correspond to their title and/or subtitles.

- Some statements should be made with caution. At the end of the manuscript, for example, you mention “The limitation of our study was the use of cross-sectional data for a causal relationship”. However, it is important to remember that causal relationships can never be made with data from a cross sectional study. Therefore, you do not have a causal relationship.

6. PLOS authors have the option to publish the peer review history of their article (what does this mean?). If published, this will include your full peer review and any attached files.

Reviewer #1: No

Reviewer #2: **Yes: **Isabel Giacomini Marques

---

## [Author Response · Author response to Decision Letter 0]

25 Jan 2023

As suggested by the editor, we have replaced the statistical analysis using Prevalence Rates with Poisson Regression.

---

## [Editor Report · Decision Letter 1]

5 Feb 2023

Determinants of stunting in children under five years old in South Sulawesi and West Sulawesi Province: 2013 and 2018 Indonesian Basic Health Survey

PONE-D-22-23326R1

Dear Dr. Anastasia,

We’re pleased to inform you that your manuscript has been judged scientifically suitable for publication and will be formally accepted for publication once it meets all outstanding technical requirements.

Kind regards,

Marly A. Cardoso, Ph.D.

Academic Editor

PLOS ONE

---

## [Editor Report · Acceptance letter]

3 May 2023

PONE-D-22-23326R1 

Determinants of stunting in children under five years old in South Sulawesi and West Sulawesi Province: 2013 and 2018 Indonesian Basic Health Survey 

Dear Dr. Anastasia:

I'm pleased to inform you that your manuscript has been deemed suitable for publication in PLOS ONE. Congratulations! Your manuscript is now with our production department. 

Kind regards, 

on behalf of

Dr. Marly A. Cardoso 

Academic Editor

PLOS ONE